# Design and Experiment of Dry-Ice Cleaning Mechanical Arm for Insulators in Substation

**Shufeng Tang \*** , **Pengfei Zhou, Xu Wang, Yue Yu and Hualei Li**

School of Mechanical Engineering, Inner Mongolia University of Technology, Hohhot 010051, China; 20171000007@imut.edu.cn (P.Z.); sting_777@163.com (X.W.); yuyueimut@163.com (Y.Y.); dpysyk@ea-c.cn (H.L.)

\* Correspondence: tangshufeng@imut.edu.cn; Tel.: +86-186-4710-3701



**Featured Application: This paper developed a kind of insulator electrified dry-ice cleaning mechanical arm, which can be used for electrified cleaning of insulator or other similar equipment in substation.**

**Abstract:** Polluted flashover of insulators has always been a crucial hidden peril to the reliability of substation for supplying electricity. At present, the insulator of the substation is mainly cleaned manually in the case of power-cut, which greatly affects the reliability of power supply and operation safety. In this paper, an insulator dry-ice mechanical arm capable of electrified working is proposed. Under the regular operation of the power grid, the operator can remotely control the mechanical arm to clean the insulators of the substation with high-pressure gas mixed with dry-ice particles. This will not only ensure the regular operation of the power grid, but also greatly protects the safety of personnel. Considering the operation environment and high-voltage working conditions of the substation, the mechanism and control system of the dry-ice cleaning mechanical arm are designed, and the motion analysis, simulation, and strength analysis of the mechanical arm are carried out. Finally, the high-voltage-insulation test of the mechanical arm and the actual operation of the substation insulator cleaning working test are carried out, which proves that this mechanical arm can adapt to the operation environment of the substation, and the working is stable, safe, and reliable, which has an important application value to solve the problem in the cleaning of the substation insulator and other electrified equipment.

**Keywords:** cleaning robot; substation insulator; dry-ice cleaning; electrified operation

## 1. Introduction

In the long-term operation of substation insulator, dust floating in the air is easy to adhere to the insulator surface to form a dirty layer, which is more serious in heavily polluted areas. In the fog, rain, snow, and other weather conditions, the insulator insulation performance is reduced, and pollution flashover accidents are easy to occur [1–4]. With the aggravation of pollution, the deterioration of global climate and environment year by year, and the use of high-voltage power grid, pollution flashover accidents show a trend of high frequency and wide area, which brings great hidden danger to power supply safely [5–8].

At present, antipollution flashover measures at home and abroad mainly include antipollution flashover organic RTV (room-temperature vulcanized) paint, electrified cleaning, and manual cleaning. There are many researches and applications of antipollution flashover RTV paint [9–14]. RTV paint can alleviate but cannot fundamentally solve the problem of insulator surface pollution, and then it also needs manual or electrified auxiliary cleaning. For the work of electrified cleaning, some substations have carried out the high-pressure water cleaning with fixed nozzles [15–18]; there are fire-fighting

or engineering vehicles modified into electrified water washing robots [19–23], which need to be driven manually, so that there are some safety risks. In other conditions, some special electrified water washing tracked robots are designed [24]. The United States, Australia, Israel, and some other countries, which adopts the helicopter water cleaning way, have passed the electrical discharge demonstration experiment in 500 kV equipotential and over-voltage. However, electrified working with helicopters is not only relatively expensive and complex, but also requires sufficient safety flight distance, so that it is difficult to achieve thorough cleaning [25–28]. The method of high-pressure water washing realizes the electrified cleaning of insulator, but the cleaning ability for industrial pollution is poor, the operation effect is not good in the area of heavy air pollution, and it also wastes water-resources greatly, so that it is not suitable for the operation in dry and cold areas. In addition, some research institutions have carried out the research of insulator brush cleaning robots [29–33], which requires high precision of robot end positioning. Due to the design of end driving and transmission mechanism, the design of high-voltage-insulation is relatively difficult, and the cleaning effect of oil-pollution is not good [34].

At present, due to the influence of cleaning effect and operation safety, the substation insulator mainly adopts the manual cleaning mode of power-cut condition (as shown in Figure 1b), however there are hundreds of insulators in each substation above 220 kV (as shown in Figure 1b), which greatly affects the safety and reliability of power supply and the intelligent construction of power grid. Therefore, it is imperative to develop a kind of electrified cleaning equipment for substation insulator.

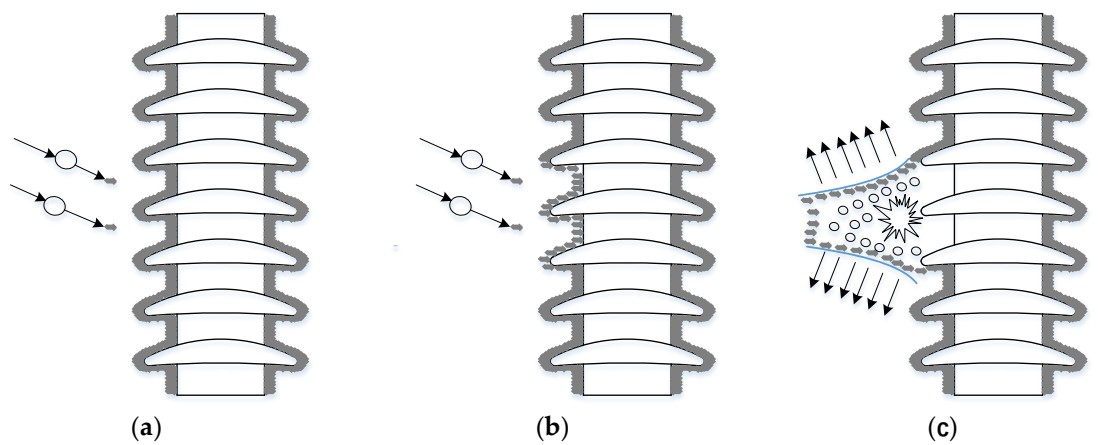

**Figure 1.** Schematic diagram of dry-ice cleaning insulator: (**a**) When dry-ice strikes the polluted layer; (**b**) when the polluted layer breaks; (**c**) when microexplosion air-flow and compressed air-flow remove the pollution.

According to this case, this paper develops an electrified dry-ice cleaning mechanical arm for high-voltage substation insulators. This system carries on electrified ice cleaning operation of substation insulators by wireless remote-control mechanical arm, and uses dry-ice particles as medium to accelerate in high-pressure air flow, impacts the wash-needed surface of insulators to clean and remove pollutions, which possesses the advantages of good cleaning effect for pollutions, nonstop operation, and high cleaning efficiency, and it can effectively solve the problem of insulator pollution flashover in substation.

## 2. General Design of Dry-Ice Cleaning Mechanical Arm System

Compared with conventional methods, dry-ice cleaning technology is more safe, economical, and nondestructive [35,36], which can effectively remove the contaminated particles on the surface of metals, insulators, and semiconductors [37]. Hoenig [38,39] thinks that the microsolids in high-speed jet can improve the removal efficiency of surface particles. The power frequency voltage and current-leakage test of dry-ice cleaning system shows that dry-ice cleaning has been widely used in on-wire cleaning of transformer, insulator, generator, and steam turbine due to its unique insulation

performance [40,41]. Due to the good insulation performance, cleaning effect, and environmental protection characteristics of dry-ice cleaning, this paper adopts the method of dry-ice cleaning insulator, using high-pressure gas carrying dry-ice particles to impact the insulator surface. When the dry-ice strikes the insulator surface, it will vaporize instantaneously with the change of thermodynamic state, and the heat exchange between the dry-ice particles and the insulator surface will occur rapidly, resulting in the rapid cooling of the surface, and the dirt will be frozen, embrittled, cracked, the bonding force with the cleaned surface will be reduced, and even happens with the "bone separation" falling off (the thermal expansion coefficient of two different materials is different). Dry-ice particles (powder) expand nearly 100 times in an instant after getting into the dirt cracks, thus causing multipoint "micro blasting" at the impact point and blowing the dirt off. At the same time, with the grinding and impact of dry-ice particles and with the blowing and cutting of compressed air, the dirt is peeled off from the cleaned surface in solid form, thus achieving the purpose of dirt removal, as shown in Figure 1.

There are at least hundreds of insulators or at most thousands of insulators in different substations, and the aerial high-voltage transmission wires crisscross, the ground high-voltage equipment is complicated, so it is difficult to use helicopters, UAVs, and other air cleaning operations to take aerial cleaning operation. Based on that, this paper proposes a kind of electrified dry-ice cleaning mechanical arm system, which is mainly composed of mobile platform, dry-ice cleaning mechanical arm, dry-ice conveying system, and hand-held controller. The mobile platform is mainly responsible for the movement of the whole system in the substation, carrying the dry-ice cleaning mechanical arm to reach the insulator needed to be cleaned. According to the cleaning task needs and the layout of the equipment in the station, different moving paths can be selected, such as spline 1 or spline 2 in Figure 2. The mechanical arm of dry-ice cleaning system drives the nozzle to move up, down, left, and right to clean the whole insulator at a fixed position, as shown in L01 to L36 in Figure 3. The dry-ice conveying system mainly includes high-pressure gas generation system and dry-ice particle conveyor. Via high-pressure gas, dry-ice particles are carried to impact the surface of insulator needed to be cleaned. The wireless communication mode is adopted between the hand-held controller and the system, which can control all direction movements of the mechanical arm and the operation for dry-ice cleaning circuit. According to the different environment of the substation, the mobile platform can be transformed by using engineering vehicles or tracked robots. This paper focuses on the research of electrified dry-ice cleaning mechanical arm.

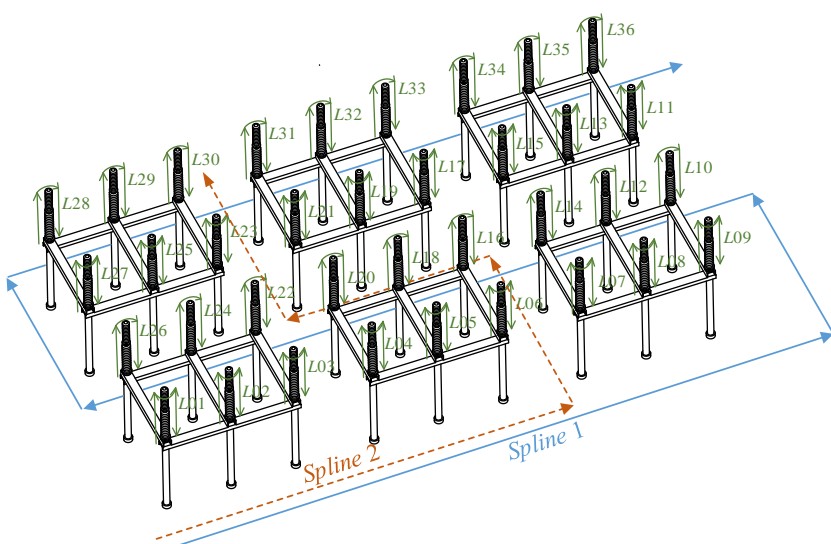

**Figure 2.** Insulator cleaning path.

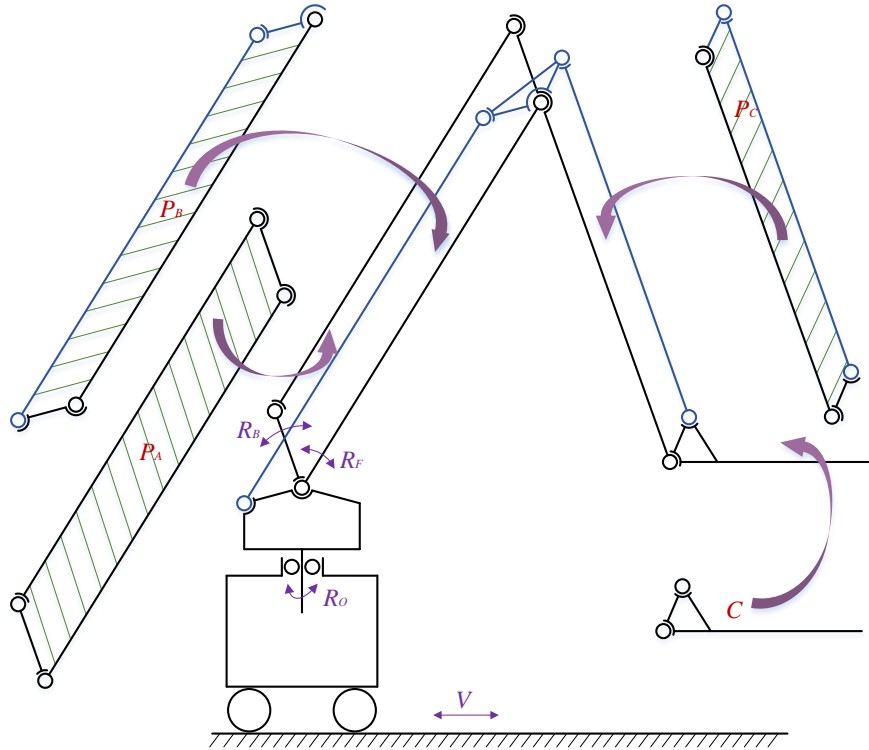

**Figure 3.** Mechanism schematic diagram of dry-ice cleaning manipulator.

In order to complete the insulator cleaning without power-cut, the dry-ice cleaning mechanical arm system must meet or be superior to the following conditions:

(1) To meet the requirements of high-voltage-insulation, the safety distance cannot be reduced due to the usage of mechanical arm;

(2) The whole system can meet the requirements of working space in the station without any interference with electrified equipment;

(3) In the process of operation, wireless control mode shall be adopted as far as possible to ensure the safety of operators;

(4) It can adapt to the high-voltage electromagnetic environment of substation and it can resist strong magnetic interference;

(5) The working space covers the whole insulator, and the cleaning operation movement is simple;

(6) Insulator cleaning has some adaptability and it can reduce the operation accuracy requirements.

## 3. Mechanical System of Electrified Dry-Ice Cleaning Robotic Arm

### 3.1. Structural Principle of Mechanical Arm

The electric dry-ice cleaning mechanical arm system adopts the combination of moving platform and mechanical arm. The mobile platform can drive the mechanical arm to reach the vicinity of the insulator. In order to adapt to the cleaning operation of the insulator pillar in the substation, the mechanical arm should be able to move forward and backward in the horizontal direction, and at the same time, it should have the lifting function in the vertical direction. On the premise of meeting the motion function, the mechanical arm shall have good high-voltage insulation characteristics, and all electrified components and conductors shall be as far as possible away from the end of the mechanical arm. Referring to the structural principle of the stacking robot, the electrified dry-ice cleaning mechanical arm uses three parallel four-bar structures $P_A$, $P_B$, and $P_C$ to form the whole body of arm (Figure 3).

Under the rotating motion of $R_B$ and $R_F$, it can drive the end frame member $C$ of the mechanical arm to move forward, backward, and up and down. Moreover, due to the characteristics of parallel four-bar linkage, no matter if the mechanical arm is in any motion state, the end frame member $C$ always maintains a fixed attitude. In order to realize the left and right movement of the mechanical arm on the horizontal plane, the root of the mechanical arm is designed with an orientation motion $R_O$, which can be combined with $R_B$ and $R_F$ to realize the front and back, left and right, and lifting movement of mechanical arm end in a certain space. Driven by the mobile platform, the whole mechanical arm can be transferred to the position where cleaning operation is required in the substation.

### 3.2. The Nozzle Design for Dry-Ice Cleaning Insulator

Considering that the diameter of the insulator is not uniform, and the lower part of the insulator is a steel frame, the top part is a high-voltage wire, so the end of the dry-ice spray will not be able to adopt the full surrounding structure. At the same time, based on the requirements of high-voltage-insulation, there should be no electrified body or conductor within the safe distance from the top electrified point, so the active actuator should be avoided at the end of the mechanical arm as far as possible. Combined with the characteristics of the constant attitude of the end components of the mechanical arm, considering the factors of insulation, safety, convenience, etc., the operation mode of up and down cross-cleaning with two side nozzles is proposed. As shown in Figure 4a, the end of the mechanical arm is composed of two cleaning circuits A and B by one spray head on both sides; under the action of direction-motion, rotating $\theta_A$ to the left side of the insulator, so the spray head on A-side forms a cleaning space $S_A$ (b); rotating $\theta_B$ to the right side of the insulator, the spray head on B-side forms a cleaning space $S_B$ (Figure 4c); the associative action of the spray heads on both sides forms a closed cleaning operation space, and the enclosed circle $S_{AB}$ is the maximum insulator cleaning range of the mechanical arm (Figure 4d). Under the action of the lifting movement of the mechanical arm, the end can complete the cleaning operation of the whole insulator pillar.

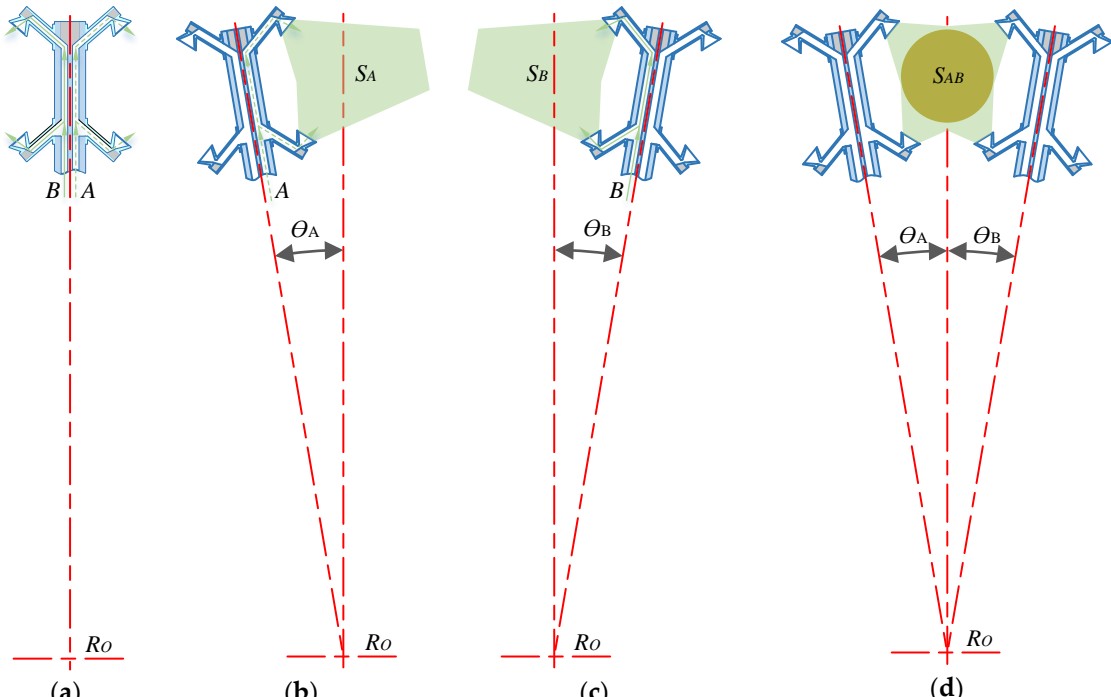

**Figure 4.** Schematic diagram of insulator cleaning nozzle and work. (**a**) Cleaning nozzle; (**b**) right nozzle cleaning area; (**c**) left nozzle cleaning area; (**d**) left and right nozzle cleaning combined area.

### 3.3. The Realization of Mechanical System of Mechanical Arm

According to the insulation coordination standard for high-voltage transmission and transformation equipment in China, the safe distance between high-voltage lead and electrified part of the equipment in the experiments is shown in Table 1 [42].

**Table 1.** The safety distance of AC(Alternating Current) and DC(Direct Current) test.

| Experimental Voltage (kV) | 50 | 100 | 200 | 500 | 750 | 1000 |
|---|---|---|---|---|---|---|
| Safe distance (m) | 1.0 | 1.2 | 1.5 | 3 | 4.5 | 7.2 |

From Table 1, it can be concluded that the safe distance required for 200 kV is 1.5 m. The mechanical arm shall not only consider the realization of movement function, but also design for considering the problem of high-voltage-insulation resistance. The safe distance to the ground for 220 kV electrified working shall not be less than 1.5 m. In the actual cleaning operation, when the end-nozzle is at the highest point of the insulator, it is closest to the electrified body. Therefore, it is required that the distance between the end and the conductive body of the robot should not be less than 1.5 m. In order to ensure the high-voltage-insulation ability of the mechanical arm, the material of MC nylon with good insulation performance, high strength, and processability is used for the arm. At the same time, the bolt material is PPS (polyphenylene sulfide) nonglass fiber or Peek (polyetheretherketone).

The three-dimensional model of the mechanical system of the mechanical arm is shown in Figure 5. The parallelogram $P_A$ is composed of the main arm, auxiliary arm, forearm, and auxiliary arm driving rod. The parallelogram $P_B$ is composed of the main arm, rear attitude linkage, attitude tripod, and arm base. The parallelogram $P_C$ is composed of the forearm, front attitude linkage, attitude tripod, and nozzle. Thus, the arm of the mechanical arm is constructed. The main arm motor drives the driving shaft of the main arm to adjust the attitude of the main arm through the synchronous belt; the auxiliary arm motor drives the driving rod of the auxiliary arm to rotate through the synchronous belt to change the shape of the parallelogram $P_A$, and then adjust the attitude of the small arm; under the combined action of the main arm motor and the auxiliary arm motor, the front and back, up and down movements of the end nozzle are realized. The parallelogram $P_B$ and $P_C$ are passive structures, keeping the attitude of the end nozzle unchanged and are always in a horizontal state. In order to balance the eccentric load moment caused by the gravity of the arm, a constant-force air spring is installed between the arm base and the main arm to reduce the driving force required by the main arm motor. The azimuth motor drives the arm base through synchronous belt to drive the whole arm to rotate horizontally relative to the mobile platform. This paper mainly studies the electrified dry-ice cleaning mechanical arm. For the convenience of the test, the mobile platform adopts two omni-directional wheels and two fixed castors to support, which can realize the omni-bearing movement of the whole mechanical arm in the substation. At the same time, the mobile platform can carry the dry-ice conveyor. The main arm motor, auxiliary arm motor, and azimuth motor are all composed of servo motor and planetary reducer. The motor, which is equipped with an incremental digital encoder, and zero position sensor compose a half-closed-loop control system.

Considering that if the external ice conveying pipeline will be hardened due to the low temperature effect of dry-ice, which will affect the operation flexibility and strength of the robot, the robot would make full use of its own structural characteristics. A double row of independent dry-ice conveying pipeline is designed in the main arm, the forearm, and the joint, so that the dry-ice can enter from the bottom of the main arm to the end of the spray gun. Since the parts of the engineering prototype are all formed by mechanical processing, some right-angle areas will be formed in the dry-ice conveying circuit, which will have some blocking effect on the transport of dry-ice particles gas and a lot of impact on the cleaning effect. The mass production of injection molding process can greatly improve the transmission effect of the ice conveying circuit.

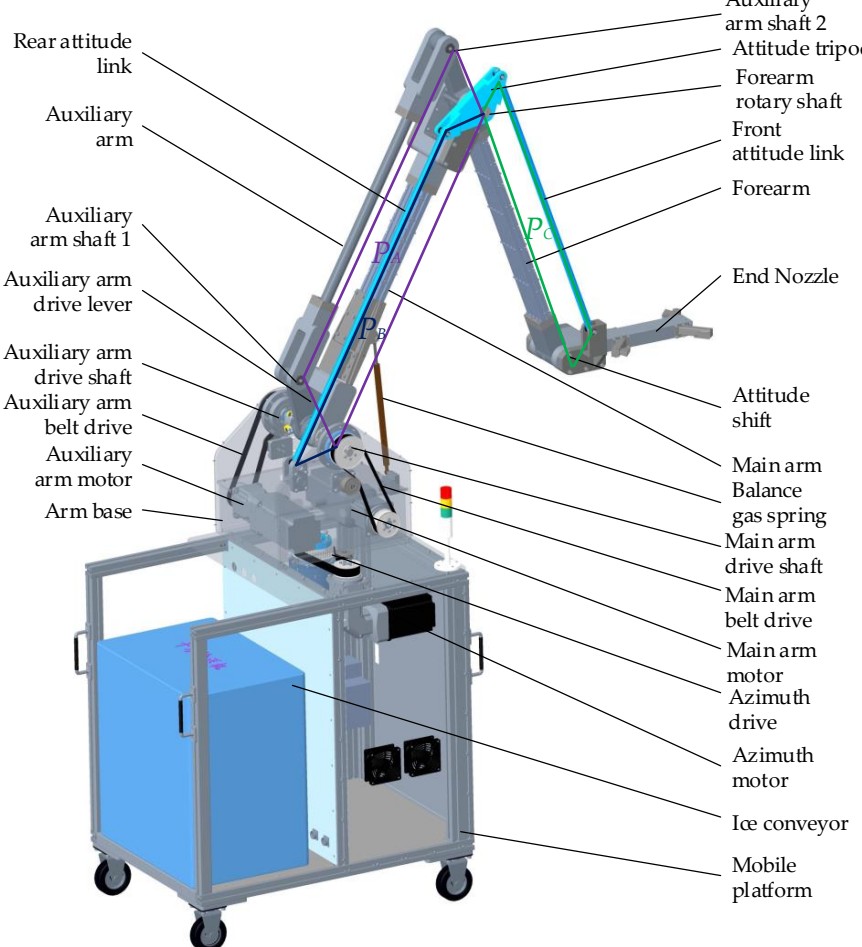

**Figure 5.** Three-dimensional model picture of arm in dry-ice cleaning robot.

The developed mechanical arm engineering prototype is shown in Figure 6. The left figure shows that the end of the arm is in a lower position when it is folded, the right figure shows that the end of the arm is in a higher state when it is lifted, and the end nozzle is always in a horizontal state.

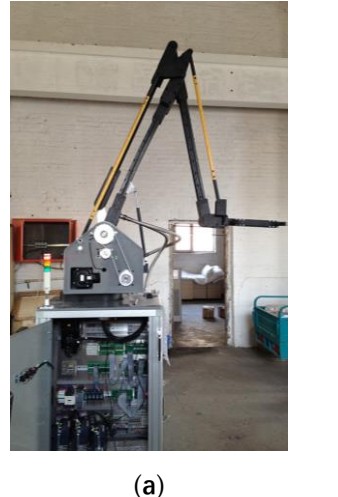

(**a**)

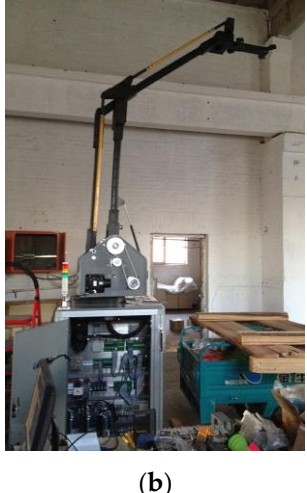

(**b**)

**Figure 6.** Dry-ice cleaning mechanical arm engineering prototype. (**a**) Mechanical arm in bottom position; (**b**) mechanical arm in top position.

## 4. Kinematics and Simulation Analysis of Dry-Ice Cleaning Mechanical Arm

### 4.1. Kinematics Analysis of Dry-Ice Cleaning Mechanical Arm

The principle of dry-ice cleaning mechanical arm is the same as that of the industrial stacking mechanical arm. Considering that the working track of the robot arm is mainly concentrated in the vertical lifting or stretching in a single plane, the schematic diagram of dry-ice cleaning mechanical arm is shown in Figure 7 [43,44]. Establish the coordinate system. The origin of the coordinate is O. The length of the main boom OQ, the connecting rod OM, the auxiliary boom MN, the small arm NP, and the end tool PT are *L*1, *L*2, *L*1, *L*3, and *L*0, respectively. The definition of the angle is negative clockwise around the Z-axis and positive counterclockwise. At the beginning, the tool at the end of the mechanical arm is at the lowest point (*Y* = 0). After a little time, the corners of the main arm and the connecting rod reach *A* and *B*, respectively.

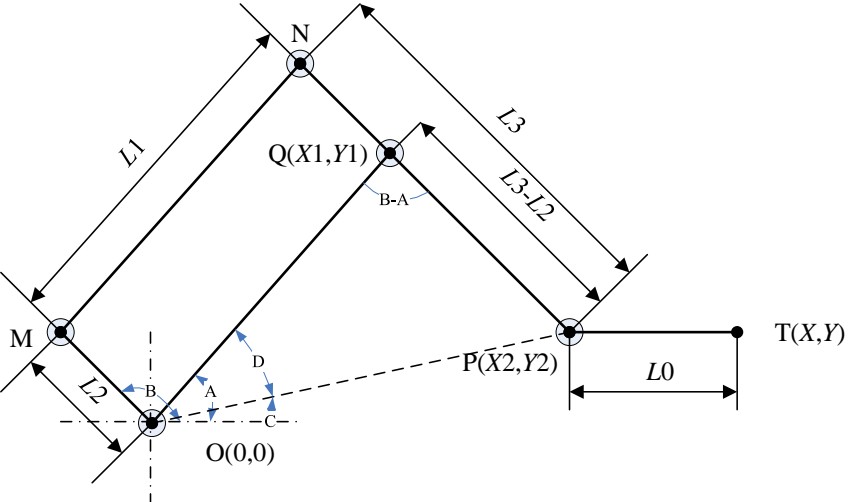

**Figure 7.** Schematic diagram of dry-ice cleaning robot.

The relationship between nozzle T (*X*, *Y*) and each rod length (*L*1, *L*2, *L*3, and *L*0) and the angle of main and auxiliary motors (*A* and *B*) is analyzed. The coordinates of each joint are: O (0, 0), A (*X*1, *Y*1), B (*X*2, *Y*2), C (*X*3, *Y*3). Among them:

1) Forward kinematics analysis:

$$\begin{cases} X1 = L1\cos A \\ Y1 = L1\sin A \end{cases} \tag{1}$$

$$\begin{cases} X2 = L1\cos A + (L3 - L2)\cos(B - \pi) \\ Y2 = L1\sin A + (L3 - L2)\sin(B - \pi) \end{cases} \tag{2}$$

$$\begin{cases} X = L1\cos A + (L3 - L2)\cos(B - \pi) + L0 \\ Y = L1\sin A + (L3 - L2)\sin(B - \pi) \end{cases} \tag{3}$$

2) Inverse kinematics analysis

As shown in the mechanism diagram 3-1, in Δ OQP:

$$\begin{cases} X2 = X - L0 \\ Y2 = Y \end{cases} \tag{4}$$

$$L1^2 + (L3 - L2)^2 - 2L1(L3 - L2)\cos(B - A) = (X - L0)^2 + Y^2 \tag{5}$$

$$\cos(B-A) = \frac{(X-L0)^2 + Y^2 - L1^2 - (L3-L2)^2}{-2L1(L3-L2)} \tag{6}$$

$$B-A = \arccos\left(\frac{(X-L0)^2 + Y^2 - L1^2 - (L3-L2)^2}{-2L1(L3-L2)}\right) \tag{7}$$

It can also be calculated:

$$L1^2 + (X-L0)^2 + Y^2 - 2L1\sqrt{(X-L0)^2 + Y^2}\cos D = (L3-L2)^2 \tag{8}$$

$$\cos D = \frac{L1^2 + (X-L0)^2 + Y^2 - (L3-L2)^2}{2L1\sqrt{(X-L0)^2 + Y^2}} \tag{9}$$

$$D = \arccos\left(\frac{L1^2 + (X-L0)^2 + Y^2 - (L3-L2)^2}{2L1\sqrt{(X-L0)^2 + Y^2}}\right) \tag{10}$$

$$\tan C = \frac{Y}{X-L0} \tag{11}$$

$$C = \arctan\left(\frac{Y}{X-L0}\right) \tag{12}$$

According to formula (5) to (12), we can get:

$$\begin{cases} A = \arccos\left(\frac{L1^2+(X-L0)^2+Y^2-(L3-L2)^2}{2L1\sqrt{(X-L0)^2+Y^2}}\right) + \arctan(\frac{Y}{X-L0}) \\ B = \arccos\left(\frac{L1^2+(X-L0)^2+Y^2-(L3-L2)^2}{2L1\sqrt{(X-L0)^2+Y^2}}\right) + \arctan(\frac{Y}{X-L0}) + \\ \quad \arccos\left(\frac{L1^2-(X-L0)^2-Y^2+(L3-L2)^2}{2L1(L3-L2)}\right) \end{cases} \tag{13}$$

The parameters in formula (13) are: $L0 = 375$ mm, $L1 = 1285$ mm, $L2 = 230$ mm, $L3 = 1310$ mm.

By establishing the kinematic equation of the dry-ice cleaning mechanical arm, the relationship between the position and pose of mechanical arm end nozzle tool and the joint (O, M, N, P, and Q) variables and the connecting rod parameters ($L1$, $L2$, and $L3$) is obtained. With the help of MATLAB mupad tool, the inverse kinematics of the mechanical arm is solved, and the motion law of each joint is calculated. Finally, the end motion track of the mechanical arm is a straight line that moves vertically up and down. The motion data of each joint is imported into ADAMS, and spline curve functions are generated, respectively.

Motion is added to the joint motion pairs of the main and auxiliary arm motors of the mechanical arm, and each motion function is changed to the corresponding spline function. Add a maker measuring point on the end nozzle for postsimulation inspection. After the simulation, the measurement results of marker point are shown in Figure 8.

It can be clearly seen from the trajectory diagram of the robot end that from 0 to 25 s, the end nozzle is always at the position 1500 mm away from the main shaft, and the trajectory is a standard straight line that from 0 to 2000 mm. The maximum rising speed of the end nozzle is 100 mm/s. The maximum azimuth velocity is 5°/s, and the simulation rotation range from 0 to −100°.

## 4.2. Simulation Analysis of Dry-Ice Cleaning Mechanical Arm

Simulation analysis of dry-ice cleaning mechanical arm with ADAMS software. The main arm and auxiliary arm are driven by 2 kW medium inertia servo motor, with a reduction ratio of 25, a transmission efficiency of 0.9, reduction ratio of synchronous belt drive of 40/28, and a transmission

efficiency of 0.9; the azimuth motor is driven by 1 kW medium inertia servo motor, with a reduction ratio of 25, a transmission efficiency of 0.9, a transmission reduction ratio of 44/28, a transmission efficiency of 0.9, and a balance gas spring of 40 kg. Considering the influence of gravity and friction comprehensively in ADAMS, the simulation is carried out according to the motor motion spline curve mentioned above, with the simulation time of 25 s and steps of 500. The acceleration, speed, and torque change curves of the three driving motors are obtained through simulation, as shown in Figure 9.

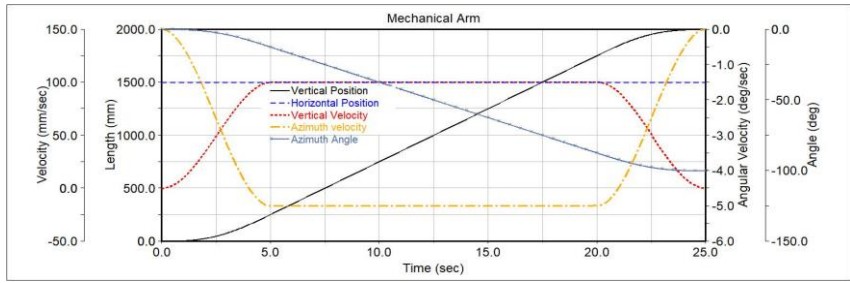

**Figure 8.** The trajectory of end-nozzle for dry-ice cleaning mechanical arm.

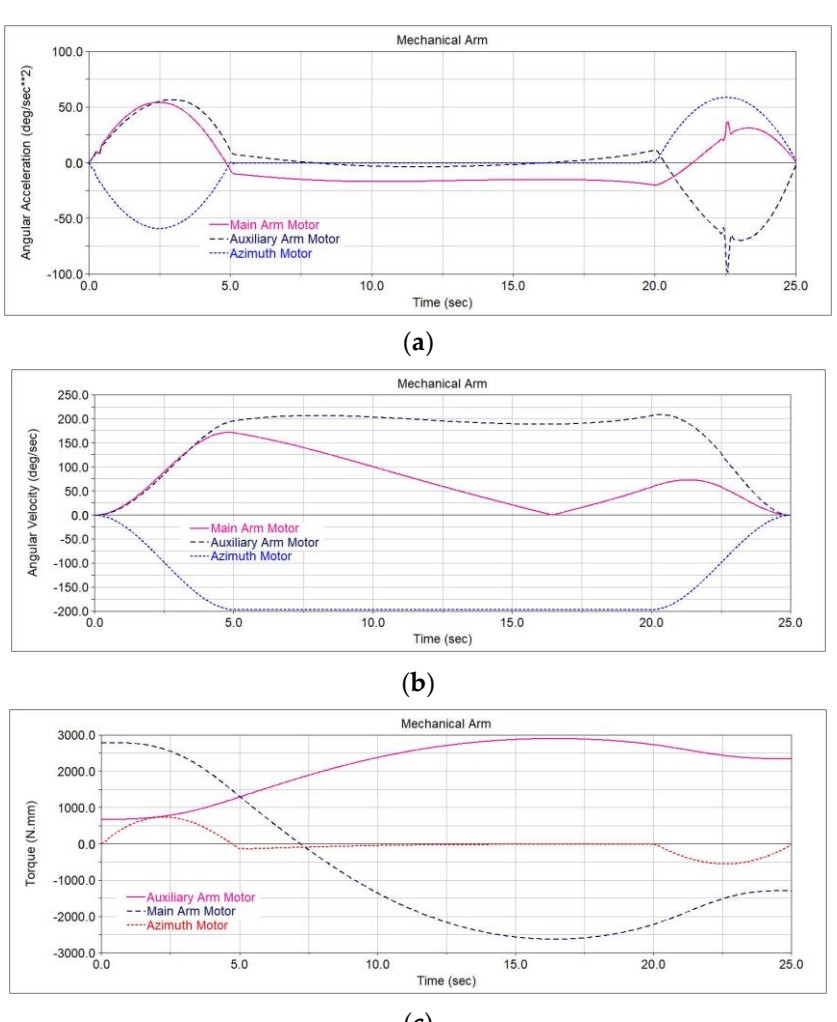

**Figure 9.** Curve of simulation results of dry-ice cleaning mechanical arm: (**a**) Acceleration curve of each shaft driving motor; (**b**) angular velocity curve of each shaft driving motor; (**c**) torque curve of each shaft driving motor.

As shown in the figure, the maximum angular velocity of all three motors shall not exceed 250°/s, i.e., 42 r/min, which is far less than the rated speed of motor 2000 r/min. The driving force of main arm motor and auxiliary arm motor are less than 3000 N.mm, which is less than the rated torque of motor 9550 N.mm; the maximum driving torque of azimuth motor is less than 1000 N.mm, which is far less than the rated torque of motor 4770 N.mm. The above simulation results have considered the acceleration moment of the mechanical arm movement, so it can be seen that the design meets the requirements and leaves sufficient margin to resist certain external interference.

## 5. Strength Analysis of Dry-Ice Cleaning Mechanical Arm

In order to adapt to the electrified working environment of the substation, the better insulation function material nylon is used for the body of the dry-ice cleaning mechanical arm, so that its strength is fairly worse than aluminum alloy and other metal materials. In order to ensure the safe and reliable operation of the mechanical arm at the requirements of strength, the overall strength of it is analyzed by using the finite element software ANSYS. Select the limited working state to analyze, when the arm extends 1.5 m, and the end-nozzle is at the lowest point and the highest point, respectively. The analysis results are shown in Figure 10. When the end-nozzle is at the lowest point, the maximum deformation of the end-nozzle is about 1.6 mm, and the maximum stress is 27.9 MPa; when it is at the highest point, the maximum deformation of the end-nozzle is about 3.6 mm, and the maximum stress is 20.4 MPa. The maximum deformation of its end is within the allowable range of operation requirements, and the maximum stress is far less than the allowable stress of nylon-66 of 60–80 MPa, so the mechanical arm meets the requirements in strength and deformation.

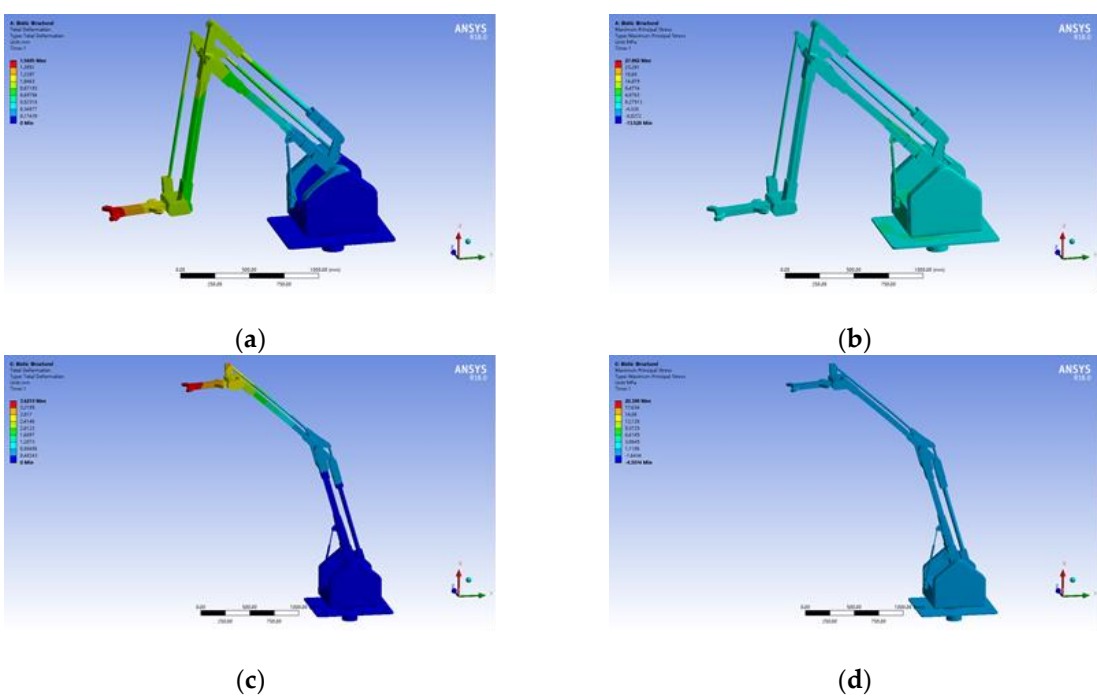

**Figure 10.** Strength analysis of dry-ice cleaning mechanical arm. (**a**) Deformation at the lowest position; (**b**) stress at the lowest position; (**c**) deformation at the highest position; (**d**) stress at the highest position.

## 6. Design of Control System

According to the requirements of substation environment and dry-ice cleaning operation, considering the safety, stability, reliability, and openness of the control system, the PMAC clipper motion control board is used as the main control unit for the control system of insulator electrified dry-ice cleaning mechanical arm, as shown in Figure 11. PMAC directly controls the main arm motor,

auxiliary arm motor, and azimuth motor and receives the feedback signal from the motor encoder, adopting the semi-closed-loop control mode; the limited-position, and zero-position signals of each motion joint are connected to the main control unit PMAC through SS11 PCB; the main control unit controls the operation of air-compressor, dry-ice conveyor, and the operation of alarm-indicator through I/O control board SS34, and at the same time, controls the ball valve to select the ice transportation loop. From the point of view of safety operation RFC100H, the mode of principal and the subordinate is adopted for wireless communication of the mechanical arm; wireless receiving module which uses the LPC1768 as the core part, and PMAC as the main control unit are developed to transmit control instructions through serial port RS232. For the convenience of operation, the hand-held controller is specially developed, and the wireless communication mode is adopted between the control system and the mechanical arm. The hand-held controller is equipped with lifting, forward and backward, rotation, cleaning operation, ice conveying circuit selection, and emergency stop buttons for mechanical arm, which can control the cleaning operation into working safely, conveniently, and accurately. In order to improve the anti-interference ability of the system for high-voltage and strong magnetism, all components of the control system are placed in a closed metal box. In order to ensure the operation safety, once the communication between the hand-held controller and the control system is wrong or interrupted, the mechanical arm cleaning operation shall be stopped immediately.

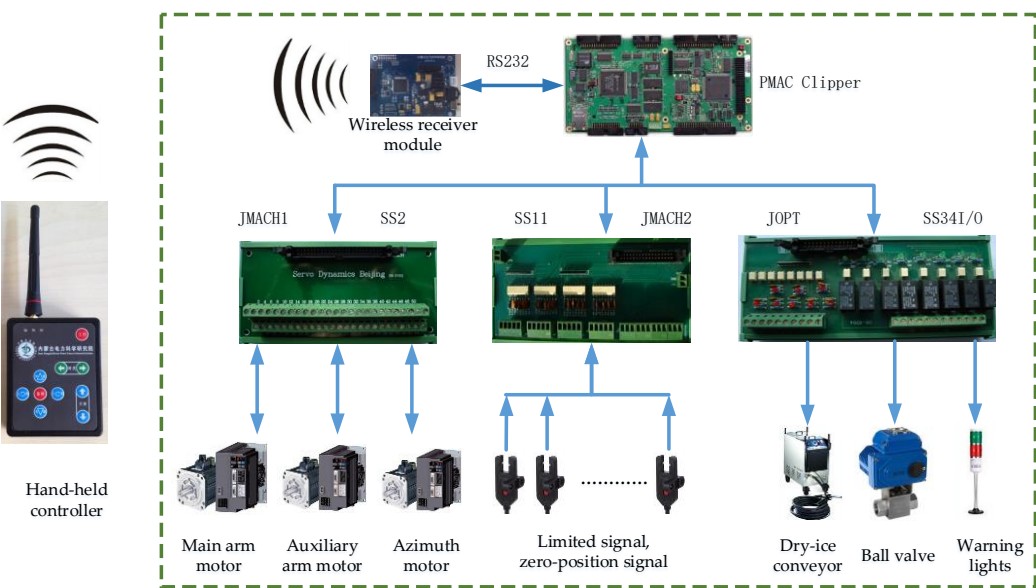

**Figure 11.** Control system of dry-ice cleaning mechanical arm.

## 7. Experiments

### 7.1. Electrified Dry-Ice Cleaning Mechanical Arm Experiment System

The electrified dry-ice cleaning robot system consists of cleaning robot, as shown in Figure 12, air compressor, gas storage tank, water-droplet separator, dry-ice conveyor, refrigerator of dry-ice, and hand-held controller. The air compressor produces compressed air and establishes a constant pressure in the storage tank. The water in the compressed gas is filtered out by a water-droplet separator to improve the air insulation characteristics. Then, the dry-ice particles are brought into the ice-translation circuit of the cleaning robot under the action of the dry-ice conveyor, and the selection of the cleaning circuit is controlled by a ball valve. The operator operates the electrified dry-ice cleaning mechanical arm through the hand-held controller at the remote-end to complete the cleaning process.

The electrified dry-ice cleaning robot uses the dry-ice particles as the medium to clean the insulator. The dry ice is very sensitive to the temperature rise and fall, and it can change the solid–liquid–gas state

in a very short time. Therefore, it is inevitable that a small amount of water molecule will adhere to its surface. With the accumulation of water particles, water droplets or water particles will be formed, which will inevitably reduce the insulation performance of the equipment. Therefore, in the whole air system, add the water droplet separator to remove the water molecules and impurities in the air flow, and minimize the effect of the water particles attached to the dry-ice particles.

The index parameters of the electrified dry-ice cleaning mechanical arm system are shown in Table 2.

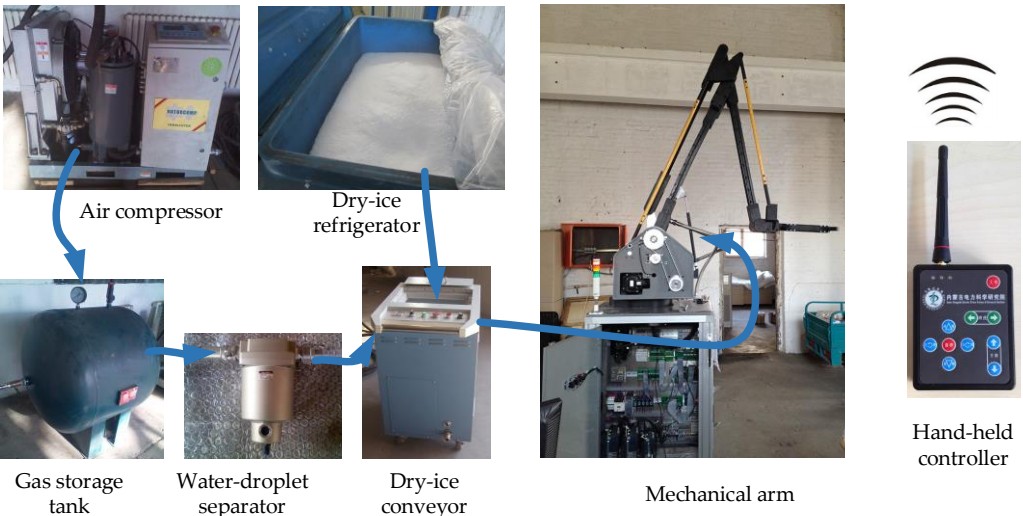

Air compressor

Dry-ice refrigerator

Gas storage tank

Water-droplet separator

Dry-ice conveyor

Mechanical arm

Hand-held controller

**Figure 12.** Dry-ice cleaning mechanical arm system.

**Table 2.** Index and parameters of mechanical arm.

| Item | Parameters | Item | Parameters |
|---|---|---|---|
| Azimuth angle range | ± 120° | Pressure of dry-ice gas | 0.5~0.6 MPa |
| Azimuth velocity | ≤ 5 °/s | Angle control accuracy | ≤ ± 0.5° |
| Working distance | 1.2~1.5 m | Terminal control accuracy | ≤ ± 5 mm |
| Working height range | 2 m | Wireless transmission distance | ≥ 100 m |
| Up, down/front, rear velocity | ≤ 100 mm/s | Wireless transmission delay | ≤ 30 ms |
| Diameter of cleaning insulator | ≤ 250 mm | Weight | ≈ 150 kg |
| Arm material | MC nylon 66 | Closing size (L × W × H) | 2000 × 800 × 2900 mm |
| Pedestal material | LY12 | Expanded maximum envelope size (Diameter × Height) | Φ 3400 × 3400 mm |

*7.2. High-Voltage Endurance Insulation Test of Dry-Ice Cleaning Mechanical Arm*

In order to ensure the operation safety of the electrified dry-ice cleaning mechanical arm in the substation and adapt to the high-voltage environment in the station (shown in Figure 13a), the 750 kV high-voltage test vehicle (shown in Figure 13b) is used for the voltage-withstanding insulation test. The test system is shown in Figure 12. After the high-voltage test vehicle is ready, the high-voltage wires from the high-voltage test transformer is directly twined on the nozzle at the end of the mechanical arm. In the process of test for the experiment, the mechanical arm is controlled by hand-held controller at a distance of 100 m to move the mechanical arm up and down, forward and backward, and rotate. At the same time, the experiment voltage is constantly raised. When the voltage is raised to 470 kV, the mechanical arm can still work stably and reliably without any phenomenon of discharge or breakdown. Through the high-voltage-insulation test, it is proved that the dry-ice cleaning mechanical arm can meet the high-voltage-insulation requirements of 220 kV substation and it can carry out electrified cleaning.

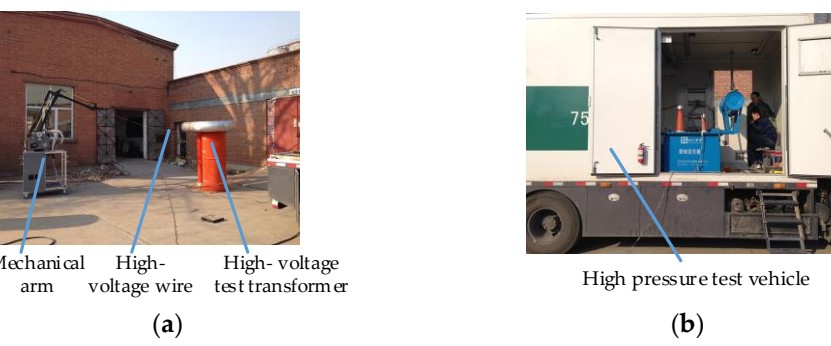

**Figure 13.** High-voltage-insulation test of dry-ice cleaning mechanical arm. (**a**) High-voltage-insulation test; (**b**) high-voltage test vehicle.

### 7.3. Insulator Cleaning Test at Substation Site

According to the requirements of electrified operation of power equipment, a cleaning test was carried out in a 220 kV substation in Hohhot. As shown in Figure 14a,b, the mechanical arm, air compressor, ice conveyor, and refrigerator transported to the substation site, and then placed the air compressor and air tank in the safety area outside the interval, leading the gas into the interval through the hose, and manually moving the mechanical arm and ice conveyor to the safety area under the insulator which needed to be cleaned. The operator controls the mechanical arm in real-time via the hand-held controller from a distance of 10 m away with the equipment, for controlling the position of the end actuator relative to the insulator through the rotary keys front, back, left, and right; controlling the end nozzle to move up and down along both sides of the insulator through the lifting key, and then completing the cleaning action; through the left and right cleaning keys, controlling the ball valve to select the dry-ice cleaning circuit at the left and right sides of the end of the mechanical arm for cleaning operation. In the process of electrified dry-ice insulator cleaning, the system is almost free from the strong electromagnetic interference of the substation, and the movement control of the mechanical arm is flexible, which can meet the needs of cleaning operation, as shown in Figure 14c,d. In the station, 12 insulator pillars were cleaned at one time. The mechanical arm and dry-ice cleaning method met the requirements of high-voltage electrified working without flashover discharge and other problems. As the insulator is in operation state, the performance test of insulator before and after cleaning is not carried out, but from the visual observation, the cleaning effect of surface-pollution is very good.

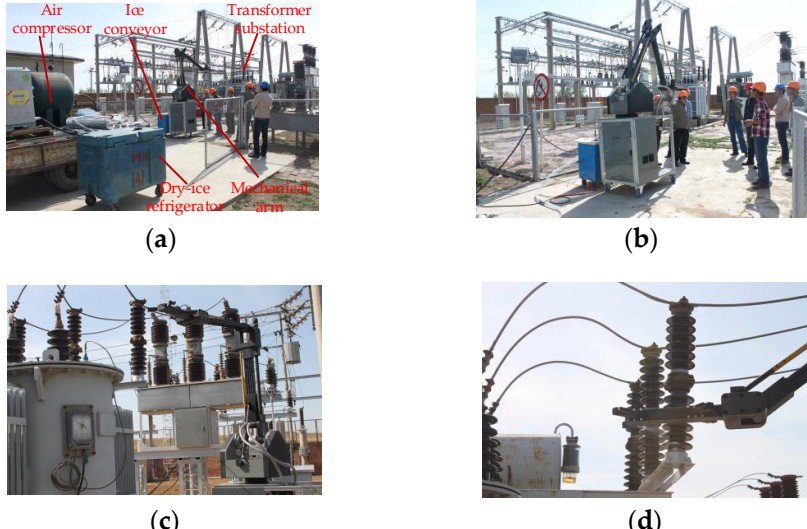

**Figure 14.** Dry-ice cleaning experiment of insulator in operation substation. (**a**) Experimental equipment; (**b**) experimental preparation; (**c**) far view of cleaning process; (**d**) close view of cleaning process.

## 8. Conclusions and Future Work

This paper presents the design and development of an electric dry-ice cleaning mechanical arm system for insulators in substations, which includes an electrified dry-ice cleaning mechanical arm, an ice conveying system, and a hand-held controller. Under the normal operation of the power grid, the remote-control arm can clean the insulator of the substation with high-pressure gas carrying dry-ice particles, which will not cause any harm to the equipment and personnel in the substation. It is proved that the cleaning effect is good and has the benefit of environmental protection by the test of the electrified dry-ice cleaning operation in the actual substation. 1) The electric dry-ice cleaning arm adopts the back-positioned driving parallel four-bar combination structure, uses high-voltage-insulation material, PMAC centralized control, and a variety of protection measures to ensure the reliable, safety, and stable electrified working of the mechanical arm; 2) the application of electrified dry-ice cleaning mechanical arm improves the safety of operators, improves the economic and social benefits of power-supply system, improves the automation level of electrified dry-ice cleaning, expands the range of electrified operation of power system, and effectively improves the safety and intelligence of power grid; 3) the environmental adaptability, protection measures, the degree of intelligence, and system-integration need to be further studied in order to expand the range of electrified working in the power system and effectively support the development of power grid intelligence.

**Author Contributions:** This paper has five authors. The authors made most of the contributions regarding conceptualization, development of theory, validation, verification of the analytical methods, discussion of the results, as well as the final manuscript. Individual contributions are as follows: Introduction, methodology, visualization, and original draft preparation, S.T., P.Z., and Y.Y.; review, editing, validation, and formal analysis, Y.Y., X.W., and H.L.; project administration, supervision, and funding acquisition, S.T. All authors have read and agreed to the published version of the manuscript.

**Funding:** This work was supported by the National Natural Science Foundation of China under grant no. 61763036.

**Conflicts of Interest:** The authors declare no conflict of interest.

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
