# Peer review of "Design and Experiment of Dry-Ice Cleaning Mechanical Arm for Insulators in Substation"

_applsci, doi:10.3390/app10072461_

Round 1

Reviewer 1 Report

change "electrified dry ice --" to "Electric dry ice -- arm" everywhere.

Abstract:

line 15, In this paper, a kind of electrified--

revise a kind of-be specific and focused

change electrified to electric (yes, mentioning again)

line 18-revise as this: "--- which will not cause any damage to the substation equipment but injury t the personnel"

need the physical details of the arm-the design parameters and the actual values.

weight

length, width, area, etc.

pressure

material

why this material?

eqns, etc. the whole details

Fig. 11, introduce a and b, to the 2 parts of the Figures and address them so and talk about it.

Same for Fig. 12, insert a, b, c, and d-for the 4 parts and talk about each of them in detail.

Conclusions:

303-revise as: this paper presents the design and development of an electric dry-ice cleaning ---

(so, need all design details/parameters/values in the body of the paper)

Reviewer 2 Report

No comments

Author Response

Response to Reviewer 2:

Dear reviewer:

Thank you very much for reviewing our paper. The full text has been carefully checked and revised.

We hope that our reply is to your satisfaction and that the paper has been improved on the basis of your comments. Thank you so much.

With kind regards,

Shufeng Tang, Pengfei Zhou , Xu Wang , Yue Yu and Hualei Li

Reviewer 3 Report

The paper presents a mechanical arm which allows cleaning under the regular operation of the power grid in substation insulators and other electrified equipment. The mechanical arm uses high pressure gar within dry ice particles.

The paper has valuable elements and will be of interest to a relative large audience. However,

ultimately, I feel there are major limitations within the work which I will to attempt to sumarize

below.

  • In introduction first paragraph should include references
  • References 1 to 8 are missing in text.
  • In line 36, authors should explain the acronim “RTV”.
  • In line 41, check references, [19-13], is wrong.
  • In lines 53 to 54, authors should include some reference.
  • I think that Figure 1 is no neccesary. Everybody can undestand only with the text.
  • In line 136, use subscripts for PA and PB.
  • In line 137, use subscripts for RB and RF.
  • In line 142, use subscripts for RO, RS and RF.
  • In line 222, please write 20.4 MPa.
  • Check references 26 and 27.

In my opinion, mechanical analysis is not enough, the analysis should include mathematical and / or graphical analysis of position, speed and acceleration.  Examples can be found in some articles:

Soriano E., Rubio H., Castejón C., García-Prada J.C. (2015) Design of a Low-Cost Manipulator Arm for Industrial Fields. In: Flores P., Viadero F. (eds) New Trends in Mechanism and Machine Science. Mechanisms and Machine Science, vol 24. Springer, Cham

Rahman, N., Carbonari, L., Caldwell, D. et al. Kinematic Analysis, Prototypation and Control of a Novel Gripper for Dexterous Applications. J Intell Robot Syst 91, 193–206 (2018).

Round 2

Reviewer 3 Report

I propose to accept the manuscript for publication